# *Escherichia coli*-Based Cell-Free Protein Synthesis for Iterative Design of Tandem-Core Virus-Like Particles

**DOI:** 10.3390/vaccines9030193

**Published:** 2021-02-25

**Authors:** Noelle Colant, Beatrice Melinek, Stefanie Frank, William Rosenberg, Daniel G. Bracewell

**Affiliations:** 1Department of Biochemical Engineering, University College London, London WC1E 6BT, UK; noelle.colant.17@ucl.ac.uk (N.C.); b.melinek@ucl.ac.uk (B.M.); stefanie.frank@ucl.ac.uk (S.F.); 2Division of Medicine, UCL Institute for Liver and Digestive Health, Royal Free Campus, London NW3 2PF, UK; w.rosenberg@ucl.ac.uk

**Keywords:** cell-free protein synthesis, virus-like particle, tandem-core, influenza vaccine

## Abstract

Tandem-core hepatitis B core antigen (HBcAg) virus-like particles (VLPs), in which two HBcAg monomers are joined together by a peptide linker, can be used to display two different antigens on the VLP surface. We produced universal influenza vaccine candidates that use this scaffold in an *Escherichia coli*-based cell-free protein synthesis (CFPS) platform. We then used the CFPS system to rapidly test modifications to the arginine-rich region typically found in wild-type HBcAg, the peptide linkers around the influenza antigen inserts, and the plasmid vector backbone to improve titer and quality. Using a minimal plasmid vector backbone designed for CFPS improved titers by at least 1.4-fold over the original constructs. When the linker lengths for the influenza inserts were more consistent in length and a greater variety of codons for glycine and serine were utilized, titers were further increased to over 70 μg/mL (4.0-fold greater than the original construct) and the presence of lower molecular weight product-related impurities was significantly reduced, although improvements in particle assembly were not seen. Furthermore, any constructs with the C-terminal arginine-rich region removed resulted in asymmetric particles of poor quality. This demonstrates the potential for CFPS as a screening platform for VLPs.

## 1. Introduction

Virus-like particles (VLPs) are self-assembling conformational epitopes that consist of viral structural proteins. These particles resemble viruses in many ways, but unlike viruses, they do not contain genetic material and therefore lack pathogenicity as well as replicative abilities. Due to their similarity in structure to their corresponding viruses, VLPs have the ability to invoke strong B cell and T cell responses even when delivered in low doses [1]. This also makes them valuable vaccine candidates; their efficacy could result in a reduction in vaccine costs because only a small amount of material is necessary to generate a strong immune response, even in the absence of adjuvants [2]. Existing vaccines on the market that utilize VLPs include Merck’s Recombivax HB and GlaxoKlineSmith’s Engerix, both for the hepatitis B virus, as well as Merck’s Gardasil and GlaxoKlineSmith’s Cervarix, which are used to treat human papilloma virus (HPV) [3,4]. Currently, over 100 VLPs for over forty different viruses, including the human immunodeficiency virus (HIV), the influenza virus, the Ebola virus, and the Zika virus, are being explored as potential vaccine candidates and have been produced in a variety of different hosts including bacteria, yeast, insect, plant, and mammalian cells [4,5,6].

Not only can VLPs be administered as vaccines to their corresponding virus, but they can be used as scaffolds to display other antigens. These epitopes are either fused genetically to VLP subunit proteins or attached by covalent or noncovalent mechanisms to the VLP surface [1]. Such VLP epitope display systems have been developed to induce a T cell response against the formation of viral tumors, to incite an immune response to a self-antigen, and to lower nicotine levels in the brain, amongst other applications [4,7]. However, some foreign antigens are too large or too hydrophobic and can cause the structural disintegration of the VLP when attached. This has led to the design of hepatitis B tandem-core VLPs as an attempt at a universal influenza vaccine.

The typical hepatitis B virus has a capsid shell composed of a surface antigen (HBsAg) with a core antigen lattice underneath (HBcAg) (Figure 1A). Serotypes *adw* and *ayw* were both used in this project. In the HBcAg VLP, 90–120 dimers each composed from individual HBcAg proteins self-assemble into an icosahedral capsid shell through trimeric or pentameric intermediates [5,6]. In hepatitis B tandem-core VLPs, two hepatitis B core proteins form a dimer and are linked by a poly-glycine–serine linker protein (Figure 1B) [8,9]. All of the VLPs used in this project contained a linker with seven repeats of the glycine–glycine–serine sequence. The linker stabilizes the dimer and assists in assembling other dimer units into a VLP [8]. The VLPs can be formed of homo-tandem dimers or hetero-tandem dimers. The homo-tandem dimers are composed of two C-terminally truncated molecules terminating at amino acid 149, while the hetero-tandem dimers have one truncated molecule and one full length molecule with a nucleic acid binding sequence extending to amino acid 185, which affects particle assembly and stability (Figure 1C) [9]. Generally, the hetero-tandem dimer constructs result in more homogenous particles than the homo-tandem dimer constructs [9]. In each dimer, there are two major insertion regions (MIRs), where antigens from various other viruses can be bound. Because the linker protein gives the particle more stability, larger and more hydrophobic antigens can be incorporated into the VLP.

For the universal influenza A vaccines expressed in this study, the influenza A matrix 2 protein ectodomain (M2e) and hemagglutinin stalk globule (HA2) have been incorporated into the MIRs. Both M2e and HA2 are similar across a range of viral strains and have been shown to provide broad protection when administered separately, against influenza A and three different influenza strains, respectively [10,11,12,13].

There are three tandem-core VLPs that are of particular interest to this project (see Figure 2). The first is the tandem-core dimer called the K1K1 VLP which contains a single amino acid, lysine, in its MIRs rather than an antigen, making the MIRs essentially “empty”. VLP 3 contains three M2e antigens from different influenza A strains in one MIR, the other MIR is empty; it does not contain a lysine. VLP 3 has been successfully produced, purified, and quantified on multiple occasions [8]. Much like VLP 3, VLP 1 contains three distinct M2e antigens in one MIR, but it also has a single HA2 antigen in the other MIR. This combination of antigens should allow for a more potent vaccine. Unfortunately, because of the number and size of antigens present in VLP 1, it has yet to be produced with the same level of success seen with VLP 3. A study of mice immunized with either VLP 1 or VLP 3 generated via cell-based processes experienced a humoral immune response and had a 100% survival rate when infected with an H1N1 influenza A virus, although the vaccine did not result in sterilizing immunity [8].

These VLPs are typically produced in *Escherichia coli* and *Pichia pastoris* expression systems [8,9]. However, cell-based production processes can take a substantial amount of time, anywhere from a few days to a few weeks, and even in high producing processes, the VLPs may have an inconsistent architecture and composition [14]. We need a more efficient system that allows us to quickly test new VLP designs. We intend to accomplish this by using an *E. coli*-based cell-free protein synthesis (CFPS) platform for screening VLPs, which should allow for the generation of higher product titers over a shorter length of time, the production of more consistently assembled particles, and the development of more scalable processes.

In CFPS methods of protein production, living cells are not engaged as part of the expression system. Instead, this in vitro production technique engages cell transcription and translation machinery to facilitate recombinant protein expression. CFPS was first used in 1961 to determine the codon for the amino acid phenylalanine [15]. Since then, it has been employed as a technique to generate a variety of therapeutic proteins, including antibodies, vaccine candidates, and protein biologics [16].

We have chosen to use *E. coli*-based CFPS as the foundation for our manufacturing scheme because it allows for flexible and fast reactions. In a traditional cell-based system, plasmid DNA containing the gene of interest would need to be incorporated into the host cell for expression. This would result in a number of clones that would need to be cultivated and compared so that the highest producing clone could be taken forward. In CFPS, there is no cloning necessary. The genetic material is simply added into the reaction. It is not necessary that this genetic material be plasmid DNA. PCR products and mRNA can also be used to express products [17,18]. This means that multiple products can be produced at the same time. There is no need to fundamentally change the other reaction components, namely, the cell extract and the concentrated reaction mixture, although some minor modifications to other process parameters may allow for dramatic improvements in titer [19].

Most CFPS reactions only take a few hours and still generate relatively high titers because the energy in the system is being used primarily for protein production rather than cell growth [20]. For example, titers of 1.3 mg/mL chloramphenicol acetyl transferase was produced with an *E. coli*-based CFPS fed-batch system in 2 h [21]. To date, the highest titer achieved in an *E. coli*-based system is 2.3 mg/mL deGFP, which was produced in batch mode in 10 h [22]. It is the simplicity of switching from one product to another and the short reaction time scales which make CFPS ideal as a screening platform.

While CFPS platforms are growing as a method of recombinant protein production, there has been relatively little work done in the realm of VLP expression via CFPS. Over a decade ago, the first *E. coli*-based CFPS platform was used to produce VLPs; MS2 bacteriophage coat protein VLPs and truncated HBcAg VLPs were expressed in 30 µL and 1 mL reaction volumes with titers of approximately 400 µg/mL in 12 h for both species [23]. More recently, human norovirus-like particles were produced in an *E. coli*-based cell free platform at roughly 600 µg/mL in 4 h, and the encephalomyocarditis virus was synthesized in a mammalian cell-based CFPS system at a rate of 1000 plaque forming units (pfu)/mL over the course of 24 h [24,25]. Here, we demonstrate that CFPS systems can also be used for the production of a more complex VLP product, the tandem-core HBcAg VLPs designed as an attempt at a universal influenza A vaccine. To the best of our knowledge, this is the first time that a CFPS system has been used to produce tandem-core VLPs. By using CFPS to express these products and novel variants of these products, we have demonstrated CFPS as a screening tool for vaccine product design.

## 2. Materials and Methods

Unless otherwise stated, all chemical reagents were purchased from Sigma Aldrich (Dorset, UK).

### 2.1. Extract Preparation

The cell extract was prepared using BL21 Star^TM^ (DE3) (Thermo Fisher Scientific, Paisley, UK) *E. coli* cells according to the protocol in Colant et al., 2020 [19]. When OD_600_ ≈ 0.6 was achieved, 500 µL of 1M IPTG was used to induced T7 RNA polymerase expression. A total protein concentration of 44 mg/mL in the cell extract was determined using a Bradford assay.

### 2.2. CFPS Reaction

CFPS reactions were performed as previously outlined in Kwon and Jewett, 2015 [26]. The cell-free reaction mixture was composed of 1.2 mM ATP, 0.85 mM each of CTP, GTP, and UTP, 1.5 mM spermidine, 1 mM putrescine, 33 mM phosphoenolpyruvate (PEP), 4 mM sodium oxalate, 0.27 mM coenzyme A (CoA), 0.33 mM nicotinamide adenine dinucleotide (NAD), 34 µg/mL folinic acid, 170 µg/mL tRNA from *E. coli* MRE 600, 90 mM potassium glutamate, 10 mM ammonium glutamate (MP Biomedicals, Eschwege, Germany), 12 mM magnesium glutamate, 1.25 mM of each amino acid except methionine, 1.5 mM methionine, 500 U/mL T7 RNA polymerase (Themo Fisher Scientific), 20% *v*/*v* extract, and 10 µg/mL plasmid (2.56 nM HBcAg, 2.36 nM K1K1, 2.31 nM VLP 3, 2.21 nM VLP 1) [26]. Reactions were performed in microcentrifuge tubes at the 100 µL scale and were agitated at 1200 rpm in an Eppendorf Thermomixer^®^ C (Stevenage, UK). The reactions were performed at 18 °C for VLP 3, which was found to improve solubility, and 30 °C for VLP 1. They proceeded for a total of 4.5 h and were then analyzed.

### 2.3. Hepatitis B Core Antigen Plasmids

All VLP plasmids were received from iQur Ltd. (London, UK). They were all designed for in vivo production.

The monomeric construct (mHBcAg) was obtained from a lab at the Latvian Biomedical Research and Study Centre which is partnered with iQur Ltd. It is a plasmid for HBcAg subtype *ayw* under the T7 promoter in a pETDuet-1 backbone. Its exact sequence is unknown, but it is ~5900 bp long. The protein sequence for the monomeric construct is shown in Appendix A.

All of the tandem-core constructs are serotype *adw*. They are in kanamycin resistant pET28b plasmids. The K1K1 VLP is a hepatitis B tandem-core protein VLP with a single lysine in each MIR; this plasmid was 6412 bp long. The protein sequence for K1K1 is found in Appendix A.

Plasmids for VLP 3, which has three matrix 2 ectodomain protein (M2e) variants in one MIR while the other MIR is empty, and VLP 1, which has a hemagglutinin stalk protein (HA2) in one MIR and three variants of M2e in the other MIR, were also prepared. These plasmids were 6637 bp and 6847 bp long, respectively. Three variations on these two vaccine candidates were designed and ordered from Genscript Biotech Corporation (Leiden, Netherlands). In one variation, the C-terminal arginine-rich region was removed; these constructs were called “VLP 3 pET28b no arg” and “VLP 1 pET28b no arg” and they were 6553 bp and 6763 bp long, respectively. Other groups have intentionally removed the arginine-rich region to improve stability and prevent lower molecular weight impurities [27]. In another variation, the original sequences were placed in the pJL1 plasmid backbone which has been designed for CFPS production. pJL1 was a gift from Michael Jewett (Addgene plasmid # 69496; http://n2t.net/addgene:69496 (accessed on 24 February 2021); RRID: Addgene_69,496). This backbone is much smaller (1766 bp) than the pET28b backbone (5209 bp) originally used for the tandem-core HBcAg VLPs. These constructs were called “VLP 3 pJL1” and “VLP 1 pJL1” and they were 3194 bp and 3404 bp long, respectively. In the final variation, the sequences were placed in the pJL1 backbone, the C-terminal arginine-rich region was removed, and the linkers around the influenza A antigens were adjusted so that they would be more uniform in length. Previously, the linkers around the hemagglutinin insert in VLP 1 were each only two amino acids long; we extended them to five and six amino acids long. The M2e inserts in the original design were placed between very long linkers which we shortened to be six amino acids long each. We also added a glycine–glycine–serine linker before the 6x histidine tag at the C-terminus of the tandem-core dimer. In addition to adjusting the length of the linkers, we also increased the variety of the codons used for the amino acids that appear more frequently in the sequences for VLP 3 and VLP 1. The original constructs were codon-optimized for *E. coli* and used nearly the same codons for each instance of glycine, serine, and histidine. We used a greater variety of codons for those three amino acids to prevent the depletion of tRNAs in the CFPS reaction. The exact changes that were made to the linkers can be seen in Appendix A. These constructs were called “VLP 3 pJL1 + linkers” and “VLP 1 pJL1 + linkers” and they were 3008 bp and 3242 bp long, respectively. Annotated versions of these sequences and the modified sequences can be found in Appendix A.

### 2.4. Hepatitis B Core Antigen Analysis

CFPS reactions expressing VLPs were analyzed via Bradford Assay, SDS-PAGE, Western blot, and dot blot.

The Quick Start^TM^ Bradford Protein Assay protocol and Quick Start^TM^ Bradford 1x Dye Reagent from Bio-Rad (Hercules, CA, USA) were used. Bovine serum albumin standards were prepared at the following concentrations: 2.0 mg/mL, 1.0 mg/mL, 0.75 mg/mL, 0.50 mg/mL, 0.25 mg/mL, 0.125 mg/mL, and 0 mg/mL. Five μL of each standard and sample were added to a single well on a clear 96-well plate. A total of 250 μL of Quick Start^TM^ Bradford 1x Dye Reagent was added to each well. The plate was incubated for 5 min at room temperature. The absorbance of each well at 595 nm was measured using a CLARIOStar^®^ Plus plate reader from BMG LabTech (Aylesbury, UK). Each standard and sample was measured in triplicate.

For SDS-PAGE, samples were with reduced with 4X SDS-PAGE sample loading buffer before being boiled at 90 °C for 10 min. The 4X SDS-PAGE sample loading buffer contains 50 mM Tris-HCl pH 6.8, 2% SDS, 10% glycerol, 1% β-mercaptoethanol, 12.5 mM EDTA, and 0.02% bromophenol blue. Samples were then applied to the lanes of a NuPAGE 12% Bis-Tris gel (Thermo Fisher Scientific) at 200 V for 50 min. The gels were then stained with InstantBlue^TM^ Coomassie Protein Stain and imaged using the GE Amersham^TM^ Imager 600 (Pittsburgh, PA, USA).

For Western blots, SDS-PAGE was performed as above, but the gels were not stained. Instead, the gels were transferred to an 8 mm × 8 mm nitrocellulose membrane using the Bio-Rad Trans-Blot^®^ Turbo^TM^ Transfer System (Hercules, CA, USA). After transfer, the membrane was stained with Ponceau S for 5 min, de-stained with Milli-Q water, and imaged. The stain was removed with tris-buffered saline (TBS). The membrane was blocked in tris-buffered saline with Tween 20 and milk powder (TBST-M) for 45 min. The membrane was incubated with the primary antibody diluted 1:1000 in TBST-M for two hours. Table 1 shows the primary antibodies used in this study. The membrane was washed with TBST three times before being incubated with the secondary antibody, an anti-mouse HRP-conjugated antibody, [HAF007] (R&D Systems, Abingdon, UK), diluted 1:1000 in TBST-M for one hour. The membrane was washed with TBST twice and TBS once. The membrane was incubated with the Thermo Fisher Scientific (Paisley, UK) Pierce ECL Western Blotting Substrate for 1–2 min in darkness then exposed and imaged using the GE Amersham^TM^ Imager 600.

For the dot blot analysis, a standard curve of recombinant HBcAg (ab49014) from Abcam (Cambridge, UK) was prepared at the following concentrations: 250 μg/mL, 125 μg/mL, 62.5 μg/mL, 31.3 μg/mL, 15.6 μg/mL, and 0 μg/mL. Then, 2 µL of each standard and sample were applied as a “dot” on an 8 mm × 8 mm nitrocellulose membrane. Standards and samples were measured in triplicate. The membrane was incubated, washed, and imaged in the same fashion as the Western blot membranes. The anti-hepatitis B virus core antigen antibody [14E11] was used for all dot blots.

### 2.5. Ammonium Sulfate Precipitation

Western blot analysis indicated that following centrifugation at 13,000× *g* for 10 min, the VLP 3 was found in the soluble fraction (supernatant) and VLP 1 was found in the insoluble fraction (pellet). The VLP 1 pellet was resuspended in renaturing buffer, 0.1 M tris buffer pH 8.7, 1 mM EDTA, 0.15 M NaCl. The soluble fraction from the VLP 3 reaction and the resuspended pellet of the VLP 1 reaction were each combined in a 1:1 ration with 1.9 M ammonium sulfate and allowed to incubate for 5 min. After the particles were pelleted by centrifugation at 13,000× *g* for 10 min, they were resuspended in renaturing buffer.

### 2.6. Transmission Electron Microscopy for Assembly Analysis

The samples were applied to a carbon/formvar-coated copper 300 mesh grids purchased from Generon (Slough, UK) for 1 min. The grid was washed with water for 5s and then negatively stained with 2% *v*/*v* uranyl acetate in water for 30s. The grids were imaged at UCL with the assistance of Mark Turmaine under a JEOL JEM-1010 transmission electron microscope (Welwyn Garden City, UK) and imaged under a Gatan Orius camera (Abingdon, UK). The grids were imaged at the University of Kent with the assistance of Ian Brown using a JEOL JEM-1230 microscope and imaged under a Gatan multiscan digital camera.

## 3. Results

### 3.1. Producing Tandem-Core VLPs in an E. coli-Based CFPS System

We used the CFPS manufacturing system to express wild-type HBcAg and three tandem-core VLPs: K1K1, VLP 1, and VLP 3 (Figure 3). The tandem-core HBcAg VLPs with the influenza antigens (VLP 3 and VLP 1) are of greatest interest to this project and were taken forward. We decreased the temperature of the reactions for both products to improve solubility. When VLP 3 was produced in reactions at 18 °C, there was less insoluble product detected (Figure 4). Although when VLP 1 was produced at temperatures below 21 °C, the material was still insoluble and very little full-length product was present following centrifugation (Figure 5). All subsequent reactions for VLP 3 and VLP 3-derived products were performed at 18 °C, and reactions for VLP 1 and VLP 1-derived products were performed at 30 °C.

### 3.2. Observing Tandem-Core Product-Related Impurities

From transmission electron microscopy (TEM) (Figure 6 and Figure 7), aggregate formation was observed for both VLP 3 and VLP 1. The aggregates may be the result of the lower molecular weight product-related impurities observed at ~37 kDa for VLP3 and ~37 kDa, ~35 kDa and ~25 kDa for VLP1 on the Western blot analysis. 

We tested CFPS reactions producing VLP 1 with four different antibodies (Figure 8). Each antibody binds to a different region on VLP 1 (Table 1). While full-length VLP 1 was detected using all the antibodies, a different product-related impurity profile was observed with each antibody. Based on the lengths of these shorter impurities, relative to the length of the full construct, the conclusion was drawn that these impurities are the result of stalled transcription or translation. A reaction with protease inhibitors supports this conclusion (Appendix B).

### 3.3. Examining the Impact of Plasmid Backbone and Gene Sequence Changes

Schematic diagrams of the modified constructs can be found in Figure 2. The two original constructs and the six modified constructs were expressed in the CFPS system under the following conditions: 20% IPTG-induced BL21 Star^TM^ (DE3) extract, complex concentrated reaction mixture pH 6.0, 5 nM of plasmid (22.6 μg/mL VLP 1, 11.2 μg/mL VLP 1 pJL1, 22.3 μg/mL VLP 1 pET28b no arg, 10.7 μg/mL VLP 1 pJL1 + linkers, 21.9 μg/mL VLP 3, 10.5 μg/mL VLP 3 pJL1, 21.6 μg/mL VLP 3 pET28b no arg, and 9.9 μg/mL VLP 3 pJL1 + linkers), 18 °C (VLP 3)/ 30 °C (VLP 1), and 4 h. They were then precipitated with ammonium sulphate (the soluble fraction was precipitated for VLP 3 and its variants and the insoluble fraction was precipitated for VLP 1 and its variants) and imaged using TEM (Figure 9 and Figure 10). They were also analyzed via SDS-PAGE, Western blot, dot blot, and Bradford assay to determine titers (Figure 11 and Figure 12). A densitometry reading of an SDS-PAGE analysis was performed for each sample (Figure 13A). The bands on the Western blot were aligned with the bands on the SDS-PAGE analysis and the density of that band was used to calculate the estimated product titer from the Bradford assay results (Figure 13B). The samples were also analyzed by dot blot and compared to a standard curve of recombinant HBcAg samples of known concentration (Figure 13C). It should be noted that the titers for the dot blot analysis are slightly higher than those estimated using densitometry because lower molecular weight product-related impurities also contribute to the signal on the dot blot analysis (Table 2).

## 4. Discussion

Both universal influenza vaccine candidates, VLP 3 and VLP 1, were largely insoluble when produced under typical conditions in the CFPS system. This is not entirely unexpected, because tandem-core VLPs produced in *E. coli* have been shown to precipitate [6]. However, it is essential that the vaccine products are soluble so that they can be easily introduced into the blood stream. Decreasing the temperature of in vivo cultures has been used to prevent the formation of insoluble inclusion bodies [28]. When we attempted to use the same strategy in our CFPS reactions, we found that there was more soluble product than insoluble product at lower temperatures (in this case, 18 °C) for VLP 3. For VLP 1, lowering the temperature of the reaction resulted in limited expression of the full-length product.

We hypothesized that VLP 3 and VLP 1 might be insoluble because they were aggregating with host cell proteins present in the cell extract. If we could remove the host cell proteins, it is possible that the tandem-core VLPs may be soluble. Because both the VLPs have a C-terminal 6x histidine tag, it should be possible to isolate them using Ni-NTA resin. Unfortunately, the particles do not bind to the resin because the 6x histidine tag is not accessible. The 6x histidine tag stabilizes HBcAg VLPs and related products because the tags join together inside of the VLP and prevent the monomer, or in this case tandem-core dimers, from separating [29]. The peptide linkers used in the tandem-core products to fuse the two monomers together also increase stability. This means that tandem-core dimers that have already come together to form particles or aggregates will not display the 6x histidine tag, as it is inside of the particle. Additionally, these particles and aggregates are very difficult to separate. Other research groups have demonstrated that tandem-core HBcAg VLPs are more resistant to denaturants, such as urea, than wild-type HBcAg VLPs [6]. It might be possible to separate the dimers by fusing another protein to tandem-core dimers that can also be used as an affinity tag, such as maltose-binding protein (MBP) or glutathione S-transferase (GST) [30]. Other proteins such as thioredoxin (TrxA), DsbC, small ubiquitin-like modifier protein (SUMO), and N-utilization substance A (NusA) can increase solubility and can be used in conjunction with smaller affinity tags, such as the 6x histidine tag [30]. These tags, which are large enough to prevent particles from self-assembling, would need to be removed following purification so that the tandem-core VLPs can properly assemble.

In the TEM images of VLP 3 and VLP 1, we see circular particles, but we also see various aggregated proteins. It is likely that many of the large proteins are residual host cell proteins from the cell extract and may be removable using a more sophisticated purification strategy. However, there may be some VLP material in the aggregate. In future studies, we could use immunogold staining to determine whether a substantial amount of that aggregated material included tandem-core VLPs [31]. Because the VLPs are insoluble, especially VLP 1, it is not entirely surprising that aggregate particles were present. In addition, when viewed under TEM, the particles formed by the VLP 1 construct were larger than anticipated, ~60 nm in diameter. Based on previous work expressing this construct in *Pichia pastoris*, we would expect to see particles that are ~40 nm in diameter [8]. The expected particle size may be achieved by adjusting the concentration of the crowding reagents used in the CFPS reaction or manipulating the agitation rate to create more protein–protein interactions [32]. It is also possible that this larger conformation is preferentially formed in CFPS reactions or proteins from the CFPS reaction which are inside of the particles or inhibiting the formation of smaller particles. Additionally, these may be aggregate species rather than symmetrical particles; a critical concentration of subunits must be present for VLPs to form properly and it may be that the amount present following the CFPS reaction is insufficient [33].

The insolubility of both products and the larger-than-expected particle size for VLP 1 may be a function of the lower molecular weight product-related impurities that were produced along with the full-length products. By examining the effects of the four different antibodies that bind to different regions of VLP 1, we were able to confirm our hypothesis that protease activity was not the cause of these impurities. If the impurities were a result of protease cleavage, we would expect to see bands of similar molecular weight, no matter which antibody was used. Instead, we saw the full-length product with all antibodies and product-related impurities with all antibodies except for [HIS.H8], the antibody corresponding to the 6x histidine tag. In fact, we only saw one band with the [HIS.H8] antibody. If the product were cleaved, we would expect to see a lower molecular weight species with the 6x histidine tag. For example, if the band we see at ~35 kDa with the [10E11] antibody, which binds to the first ten amino acids of HBcAg indicating that lower molecular weight species has a region corresponding to the beginning of the HBcAg protein, was the result of a cleavage product, then we would also expect a band at ~25 kDa with the [HIS.H8] antibody because that would be the other half of the product. Because there were no other bands on the Western blot with the [HIS.H8] antibody, this indicates that the impurities are shorter versions of the full-length product, rather than cleavage products. As a further confirmation, we also attempted CFPS reactions producing VLP 3 with a protease inhibitor cocktail added to the reaction; this showed no difference in the truncations produced (Appendix B).

Once we determined that the impurities were likely not caused by protease cleavage, we hypothesized that the lower molecular weight product-related impurities were a result of partial transcription of the plasmid DNA or partial translation or degradation of the mRNA. We estimated the size of lower molecular weight product-related impurities observed using the [14E11] antibody that corresponds to amino acids 135–140 of HBcAg (~37 kDa for VLP 3 and ~37 kDa, ~35 kDa, and ~25 kDa for VLP 1) and mapped them out, operating under the assumption that all impurities begin at the first start codon after the ribosome binding sequence in the plasmid (Figure 14). We believe that the MIR of the second HBcAg dimer is a particularly troublesome region and that the ~37 kDa fragment for VLP 3 and the ~37 kDa and ~35 kDa fragments for VLP 1 are the result of abortive transcription or translation in that region. The ~25 kDa fragment for VLP 1 is most likely the result of issues with the peptide linker holding the two HBcAg monomers together, although it is surprising that we did not see a similar fragment for VLP 3. Abortive transcripts are not uncommon in systems that utilize the T7 RNA polymerase to transcribe products that are expressed under the T7 promoter, although these mRNA sequences are typically only a few nucleotides long [34]. It is also possible that one of the essential components for translation, such as tRNAs or ribosomes, was limited in our system [35]. However, we know that these lower molecular weight product-related impurities are present in vivo in *P. pastoris* as well as *E. coli*, which may indicate that the impurities were a result of the design of the plasmid containing the gene for the product or the product itself [36]. With that in mind, we designed modified versions of the plasmids and the gene sequences for both VLP 3 and VLP 1.

Much like the original VLP 1 construct, the new VLP 1 variants were insoluble. However, modifying the constructs did have a few advantages. Changing the plasmid backbone to one that has been optimized for CFPS expression (pJL1) had no noticeable effect on the product-related impurities, but it did result in an increase in titer. Titers for “VLP 1 pJL1” were at least 1.8-fold greater than VLP 1, and titers for “VLP 3 pJL1” were at least 1.4-fold greater than VLP 3. As we saw previously, VLP 1 and “VLP 1 pJL1” form particles that are larger than anticipated. Much like with VLP 1 it is possible that different conformations are formed preferentially in CFPS reactions than in in vivo cultivations. The different morphology might also be a function of product concentration; under a certain threshold, aggregation may be favored.

In the set of reactions shown in Figure 8, there do not appear to be any lower molecular weight product-related impurities for “VLP 3 pET28b no arg” or VLP 3, but the bands are very faint, indicating that expression was low in general. Lower molecular weight product-related impurities were still present in reactions expressing “VLP 1 pET28b no arg”, but on the Western blot analysis, those bands appeared fainter compared to the bands for the full-length product. This would suggest that removing the arginine-rich region may improve the ratio of full-length product to lower molecular weight product-related impurities. Removing the arginine-rich region also increases the titer; titers for “VLP 1 pET28b no arg” were at least 2.5-fold times higher than titers for VLP 1 and titers for “VLP 3 pET28b no arg” were at least 1.3-fold higher than titers for VLP 3. However, we were not able to obtain titers for “VLP 3 pET28b no arg” using our dot blot analysis. The primary antibody used for the Western blot, [14E11], corresponds to amino acids 135–141 on the HBcAg monomer. The arginine-rich region begins at amino acid 149 in the HBcAg monomer. It is possible that by removing the arginine-rich region, the folding of the protein has changed such that the amino acids corresponding to the [14E11] antibody are less accessible. This would also explain the rather faint band we saw on the Western blot analysis of “VLP 3 pET28b no arg”. We might expect to see that same issue with “VLP 1 pET28b no arg” because the arginine-rich region is removed in that construct as well. Then again, “VLP 1 pET28b no arg” has inserts in both of its MIRs while “VLP 3 pET28b no arg” only has inserts in the first MIR, and that distinction may account for the difference in detection with each construct. Additionally, the particles formed from the constructs without the arginine-rich region appeared ill-formed and aggregated. This aligns with the results of in vivo expression in *E. coli* of tandem-core HBcAg constructs without the arginine-rich region [9].

The “VLP 1 pJL1 + linkers” and “VLP 3 pJL1 + linkers” constructs resulted in samples with little to no detectable lower molecular weight product-related impurities. This would indicate that shortening the linkers around the influenza inserts and increasing the variety of tRNAs utilized for more heavily used amino acids like glycine and serine prevents abortive transcription and/or translation that results in those impurities, thus increasing the ratio of full-length product to lower molecular weight product-related impurities. These constructs also resulted in dramatically improved titers; the “VLP 1 pJL1 + linkers” construct produced at least 4.0-fold as much product as the original construct and the “VLP 3 pJL1 + linkers” construct produced at least 3.4-fold as much. The particles from the “VLP 1 pJL1 + linkers” and “VLP 3 pJL1 + linkers” constructs were very inconsistent; some particles were symmetrical, and some were aggregated or only partially formed. This is likely the due to the removal of the arginine-rich region as we saw with the “VLP 1 pET28b no arg” and “VLP 3 pET28b no arg” constructs, although the influenza inserts may also be closer together once the linkers were shorter, introducing steric hindrance that could result in poor particle formation.

Using CFPS, we were able to simultaneously screen three new designs for each of our vaccine candidates and determine which modifications had the greatest impact on the titer, particle formation, and presence of lower molecular weight product-related impurities. This process only took two working days: one for the reactions and the ammonium sulfate precipitation and one for the analysis. In the future, this strategy could be used to rapidly iterate on the design of new VLP-based vaccines to improve product quality.

## 5. Conclusions

Using our CFPS manufacturing system, we were able to produce two universal influenza vaccine candidates, VLP 3 and VLP 1, and rapidly screen novel modified versions of these candidates. By decreasing the temperature of the reactions producing VLP 3, we were able to improve the solubility of the product. Decreasing the temperature in the reaction producing VLP 1 did not have the same effect—it remained insoluble. While we were successful in producing the desired full-length product, we also produced lower molecular weight product-related impurities. By examining the variations in Western blots performed with different primary antibodies that correspond to different parts of the tandem-core VLP protein, we were able to determine that these impurities were not the result of protease cleavage. We believe that they are either partially transcribed or partially translated species. Further work quantifying mRNA, tRNA, and ribosomes will need to be conducted to establish whether this is an issue with transcription, translation, or both processes.

Using our CFPS platform, we were able to quickly express and analyze three modified versions of the two vaccine candidates. We determined that using an optimized plasmid backbone improves titers but does not impact lower molecular weight product-related impurities. Removing the C-terminal arginine-rich region may decrease the presence of lower molecular weight product-related impurities but results in ill-formed particles, which is also true in in vivo expression of tandem-core VLPs [9]. Optimizing the backbone, removing the arginine-rich region, and adjusting the linkers around the influenza inserts significantly decreases the presence of lower molecular weight product-related impurities and improves titers but does not result in consistently symmetrical particles. Ultimately, we were able to substantially increase our understanding of these vaccine candidates and establish which changes in their sequence design increased titers, impacted assembly, and decreased the presence of lower molecular weight product-related impurities. Although it was not our goal to optimize these vaccine candidates, one might infer from these results that modified constructs of VLP 1 and VLP 3 in the pJL1 backbone with the arginine-rich region present and with the adjusted linkers should have fewer assembly issues, fewer lower molecular weight product-related impurities, and increased titers. By expressing tandem-core HBcAg VLPs in our CFPS system and designing and expressing modified versions of those VLPs, we have demonstrated that our CFPS system is not only capable of generating complex self-assembling products for vaccine production, but it is a useful screening tool for the iteration and improvement of vaccine product design.

## Figures and Tables

**Figure 1 vaccines-09-00193-f001:**
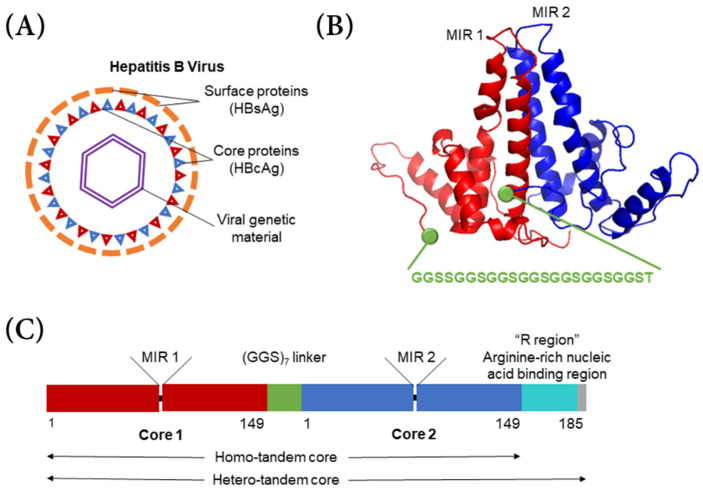
Tandem-core hepatitis B antigen structure. (**A**) Hepatitis B virus with surface antigen (HBsAg) in orange, core protein (HBcAg) in red and blue, and viral DNA in purple; (**B**) HBcAg tandem-core dimer with glycine–glycine–serine linker shown in green and major insertion regions (MIRs) labelled; (**C**) Difference between homo-tandem core and hetero-tandem core, modified from Peyret et al.

**Figure 2 vaccines-09-00193-f002:**
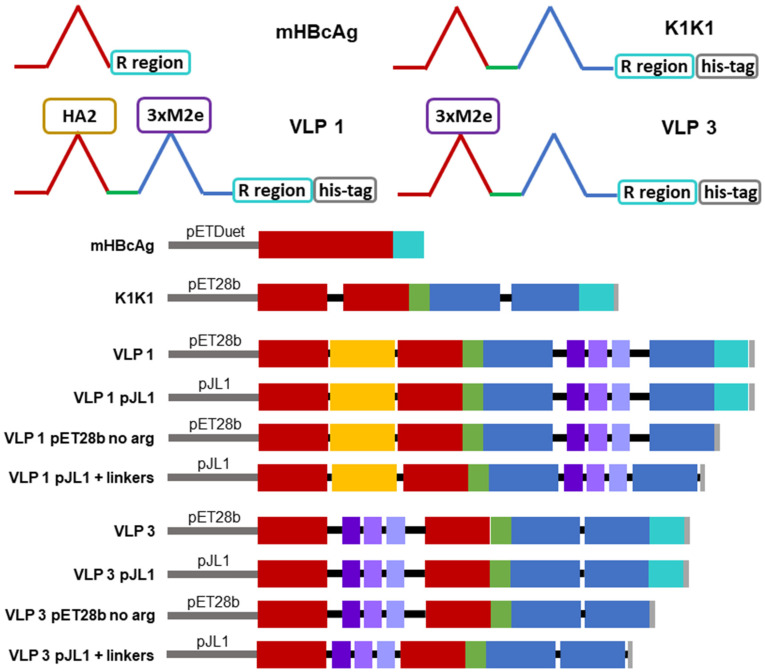
Graphical depiction of tandem-core constructs. The plasmid backbones are represented as a grey line, the first HBcAg monomer is represented with red boxes, the second HBcAg monomer is represented with blue boxes, the hemagglutinin stalk protein is represented as a yellow box, the three M2e proteins are represented as purple boxes, the arginine-rich region is represented as an aqua box, the 6x histidine tag is represented as a grey box, the linker holding the two monomers together is represented as a green box, and all other linkers are represented as black lines.

**Figure 3 vaccines-09-00193-f003:**
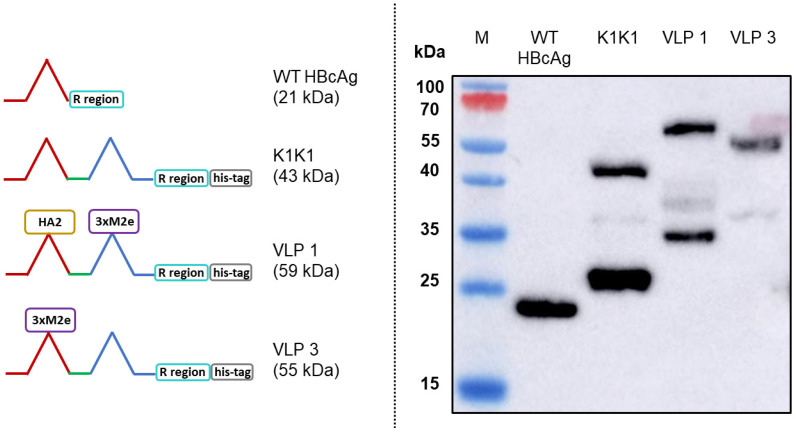
CFPS Expression of tandem-core HBcAg VLP constructs. The wild-type hepatitis B core antigen (WT HBcAg) along with three tandem-core constructs (where two HBcAg monomers were covalently joined using a glycine–serine linker) were successfully expressed in the CFPS system. The tandem-core constructs were K1K1, which has lysine residues in both of its MIRs, VLP 1, which has hemagglutinin stalk protein HA2 in its first MIR and three M2e proteins in its second MIR, and VLP 3, which has three M2e proteins in its first MIR. A Western blot with the [14E11] primary antibody was performed on all four proteins.

**Figure 4 vaccines-09-00193-f004:**
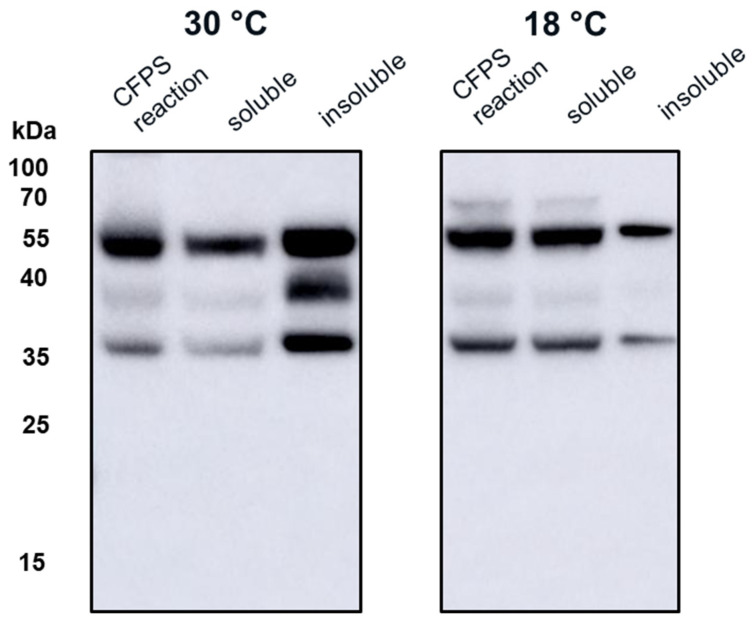
CFPS reactions to produce VLP 3 at 30 °C and 18 °C. CFPS reactions to produce VLP 3 were performed at both 30 °C and 18 °C. These reactions were centrifuged at 13,000 rpm for 10 min. The supernatant was decanted and saved as the soluble fraction. The pellet was resuspended and saved as the insoluble fraction. A Western blot with the [10E11] primary antibody was performed on the CFPS reaction, the soluble fraction, and the insoluble fraction. At 30 °C, the material in the reaction was mostly insoluble. At 18 °C, more VLP 3 could be found in the soluble fraction.

**Figure 5 vaccines-09-00193-f005:**
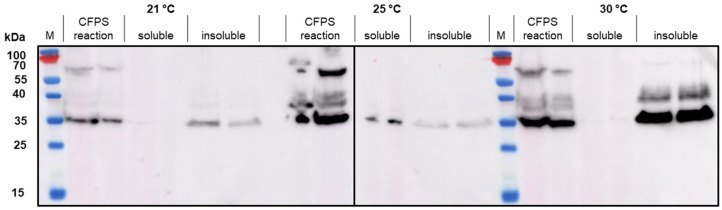
CFPS reactions to produce VLP 1 at 21 °C, 25 °C and 30 °C. Duplicate CFPS reactions to produce VLP 1 were performed at 21 °C, 25 °C, and 30 °C. These reactions were centrifuged at 13,000 rpm for 10 min. The supernatant was decanted and saved as the soluble fraction. The pellet was resuspended and saved as the insoluble fraction. A Western blot with the [10E11] primary antibody was performed on the CFPS reaction, the soluble fraction, and the insoluble fraction. At all three temperatures, the VLP 1 material was largely insoluble. At 21 °C, there was barely any full-length material in the reaction.

**Figure 6 vaccines-09-00193-f006:**
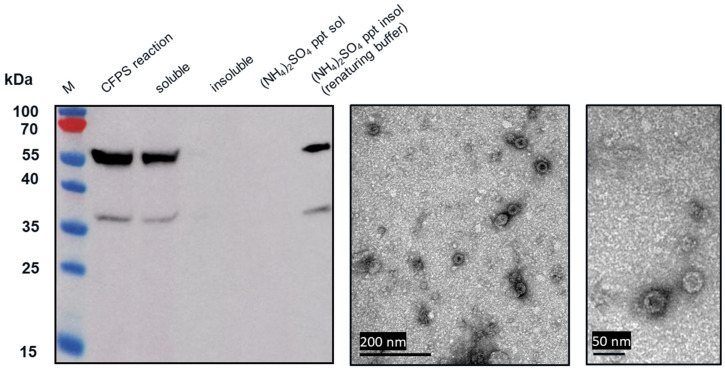
Ammonium sulphate precipitation and TEM images of VLP 3. CFPS reactions to produce VLP 3 were performed at 18 °C. These reactions were centrifuged at 13,000 rpm for 10 min. The supernatant was decanted and saved as the soluble fraction. The pellet was resuspended and saved as the insoluble fraction. The soluble fraction was then precipitated with ammonium sulphate. The supernatant was collected and saved as (NH_4_)_2_SO_4_ ppt sol. The pellet was resuspended in renaturing buffer and saved as (NH_4_)_2_SO_4_ ppt insol. A Western blot with the [14E11] primary antibody was performed on the CFPS reaction, the soluble fraction, the insoluble fraction, (NH_4_)_2_SO_4_ ppt sol, and (NH_4_)_2_SO_4_ ppt insol. (NH_4_)_2_SO_4_ ppt insol was imaged under a TEM. Symmetrical particles of ~40 nm in size are visible.

**Figure 7 vaccines-09-00193-f007:**
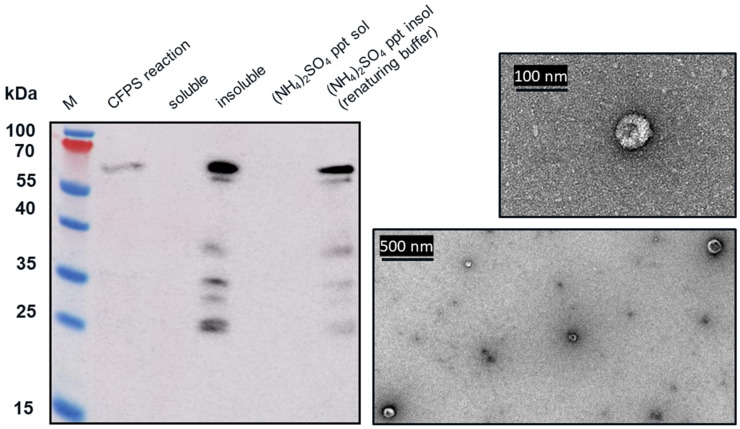
Ammonium sulphate precipitation and TEM images of VLP 1. CFPS reactions to produce VLP 1 were performed at 30 °C. These reactions were centrifuged at 13,000 rpm for 10 min. The supernatant was decanted and saved as the soluble fraction. The pellet was resuspended and saved as the insoluble fraction. The insoluble fraction was then precipitated with ammonium sulphate. The supernatant was collected and saved as (NH_4_)_2_SO_4_ ppt sol. The pellet was resuspended in renaturing buffer and saved as (NH_4_)_2_SO_4_ ppt insol. A Western blot with the [14E11] primary antibody was performed on the CFPS reaction, the soluble fraction, the insoluble fraction, (NH_4_)_2_SO_4_ ppt sol, and (NH_4_)_2_SO_4_ ppt insol. (NH_4_)_2_SO_4_ ppt insol was imaged under TEM. Symmetrical particles of ~60 nm in size are visible.

**Figure 8 vaccines-09-00193-f008:**
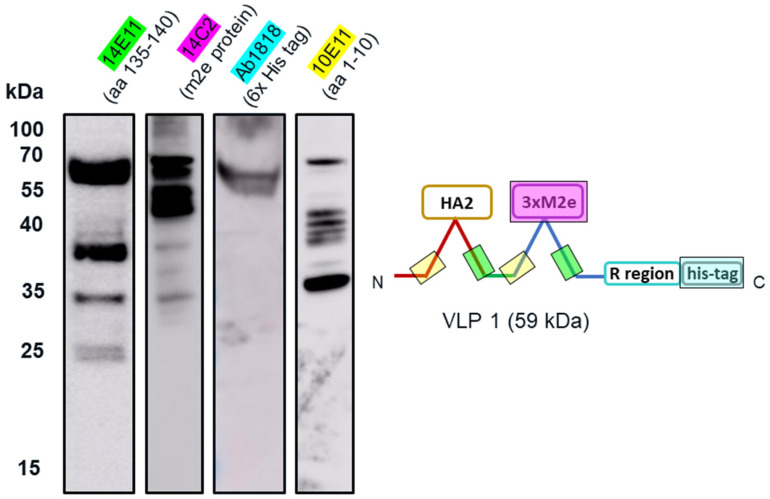
Western blot analysis of CFPS reaction to produce VLP 1 using four different primary antibodies. CFPS reactions to produce VLP 1 were performed at 30 °C. A Western blot was performed on the reactions using Anti-Hepatitis B Virus Core Antigen antibody [14E11] (ab8638), which corresponds to amino acids 135–141 on the HBcAg monomer; Anti-Influenza A Virus M2 Protein antibody [14C2] (ab5416), which corresponds to the N-terminal of the Influenza A Virus M2 Protein; Anti-6X His tag^®^ antibody [HIS.H8] (ab18184), which corresponds to any 6x histidine tag; and Anti-Hepatitis B Virus Core Antigen antibody [10E11] (ab8639), which corresponds to amino acids 1–10 on the HBcAg monomer. While full-length VLP 1 was detected using all the antibodies, a different product-related impurity profile was observed with each antibody.

**Figure 9 vaccines-09-00193-f009:**
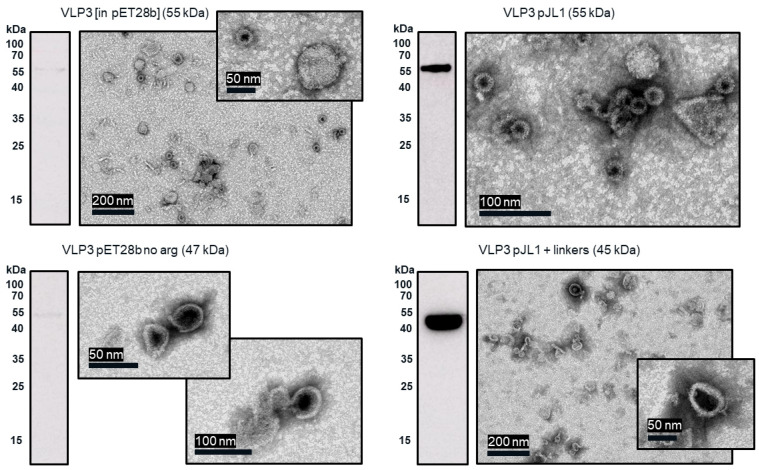
Ammonium sulphate precipitation and TEM images of VLP 3 variants. CFPS reactions to produce VLP 3 variants were performed at 18 °C. These reactions were centrifuged at 13,000 rpm for 10 min. The supernatant was decanted and saved as the soluble fraction. The pellet was resuspended and saved as the insoluble fraction. The soluble fraction was then precipitated with ammonium sulphate. The resuspended precipitate was analyzed via Western blot with the [14E11] primary antibody and imaged under TEM.

**Figure 10 vaccines-09-00193-f010:**
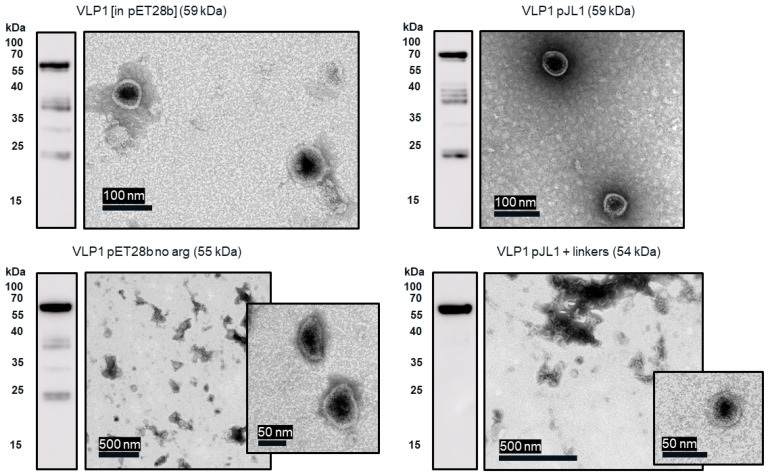
Ammonium sulphate precipitation and TEM images of VLP 1 variants. CFPS reactions to produce VLP 1 variants were performed at 30 °C. These reactions were centrifuged at 13,000 rpm for 10. min. The supernatant was decanted and saved as the soluble fraction. The pellet was resuspended and saved as the insoluble fraction. The insoluble fraction was then precipitated with ammonium sulphate. The resuspended precipitate was analyzed via Western blot with the [14E11] primary antibody and imaged under TEM.

**Figure 11 vaccines-09-00193-f011:**
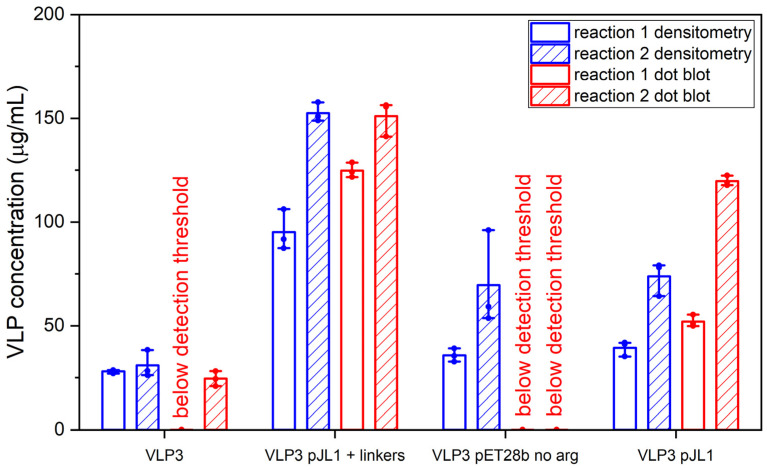
Densitometry and dot blot analysis to determine titer for VLP 3 variants. Resuspended precipitates from the CFPS reactions producing VLP 3 variants were analyzed using a densitometry reading of an SDS-PAGE analysis and a dot blot analysis for each sample. Two sets of reactions were performed (reactions 1 and 2). Error bars represent plus or minus one standard deviation for n = 3 technical replicates, each represented as a single data point.

**Figure 12 vaccines-09-00193-f012:**
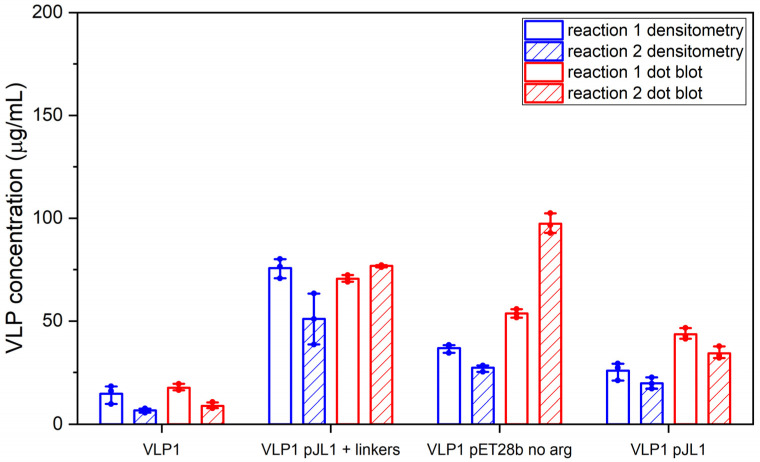
Densitometry and dot blot analysis to determine titer for VLP 1 variants. Resuspended precipitates from the CFPS reactions producing VLP 1 variants were analyzed using a densitometry reading of an SDS-PAGE analysis and a dot blot analysis for each sample. Two sets of reactions were performed (reactions 1 and 2). Error bars represent plus or minus one standard deviation for n = 3 technical replicates, each represented as a single data point.

**Figure 13 vaccines-09-00193-f013:**
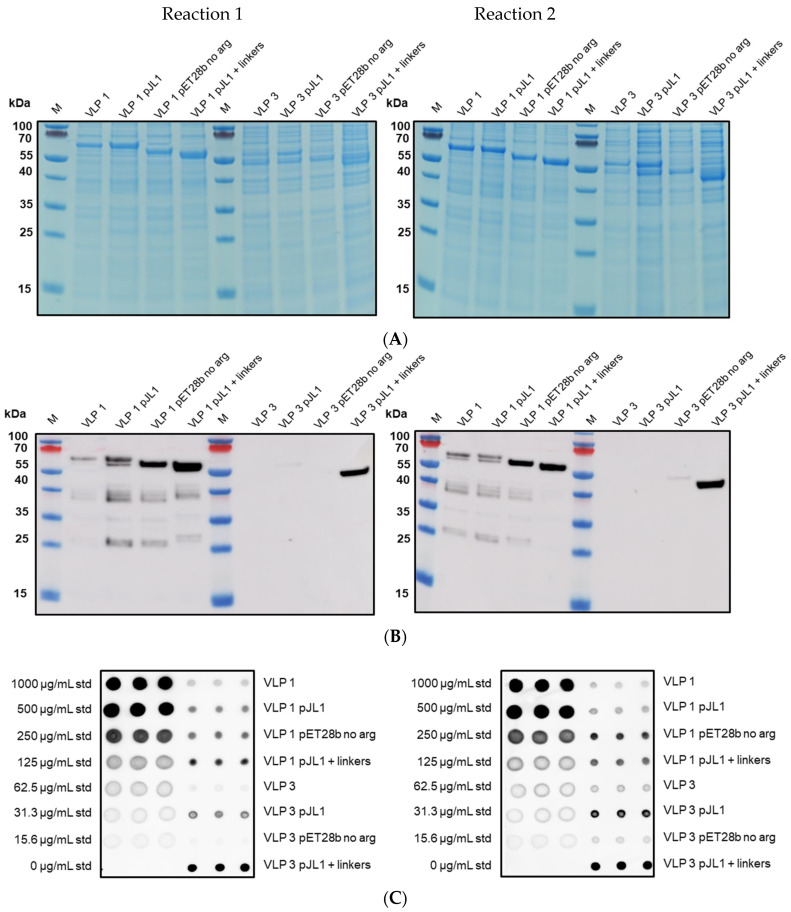
Densitometry and dot blot analysis images of VLP variants. SDS-PAGE images (**A**) were analyzed using ImageQuant software to determine the density of each band in each lane. The SDS-PAGE images were aligned with the Western blot images (**B**) to determine which band had the full-length product. The density of that band within the lane was used to calculate titers from the Bradford assay results. Dot blots (**C**) were performed with the [14E11] primary antibody and the images were analyzed using ImageQuant software. The titers of the samples (shown on the right-hand side) were calculated based on the standard curve (left-hand side).

**Figure 14 vaccines-09-00193-f014:**
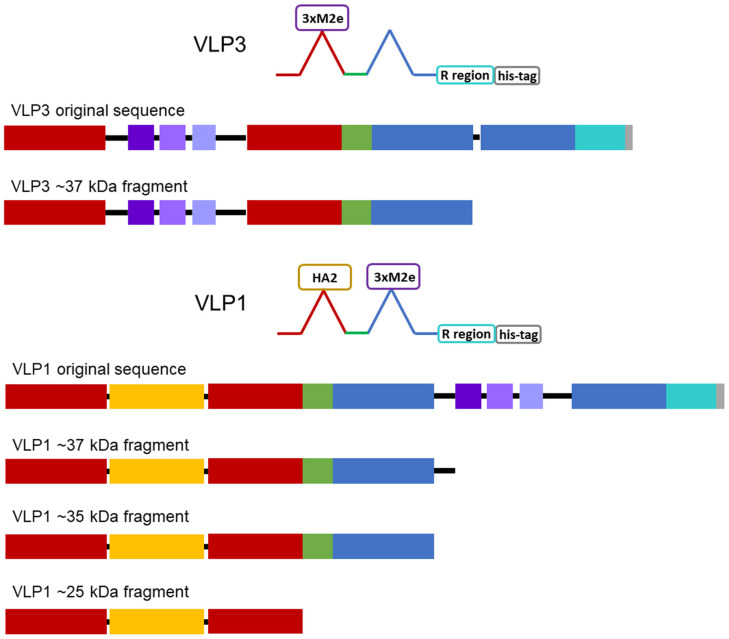
Lower molecular weight product-related impurities for tandem-core HBcAg VLPs. The first HBcAg monomer is represented with red boxes, the second HBcAg monomer is represented with blue boxes, the hemagglutinin protein is represented as a yellow box, the three M2e proteins are represented as purple boxes, the arginine-rich region is represented as an aqua box, the 6x histidine tag is represented as a grey box, the linker holding the two monomers together is represented as a green box, and all other linkers are represented as black lines. Based on the bands visible on the Western blot analyses, VLP 3 has one lower molecular weight product-related impurity, ~37 kDa in size. This fragment could be the result of abortive transcription or translation in the MIR of the second HBcAg monomer. VLP 1 has three lower molecular weight product-related impurities, ~37 kDa, ~35 kDa, and ~25 kDa in size. The ~37 kDa fragment likely extends only to the linker holding the M2e proteins in the MIR of the second HBcAg monomer, the ~35 kDa fragment likely extends only to the beginning of the MIR of the second HBcAg monomer, and the ~25 kDa fragment contains only the first HBcAg monomer with the HA2 protein in its MIR.

**Table 1 vaccines-09-00193-t001:** Mouse primary antibodies for Western blots ^1^.

Antibody Name	Corresponding Region
Anti-Hepatitis B Virus Core Antigen antibody [10E11] (ab8639)	Amino acids 1–10 on HBcAg
Anti-Hepatitis B Virus Core Antigen antibody [14E11] (ab8638)	Amino acids 135–141 on HBcAg
Anti-Influenza A Virus M2 Protein antibody [14C2] (ab5416)	N-terminal of the Influenza A Virus M2 Protein
Anti-6X His tag^®^ antibody [HIS.H8] (ab18184)	Any 6x histidine tag

^1^ All antibodies are from Abcam (Cambridge, UK).

**Table 2 vaccines-09-00193-t002:** Densitometry and dot blot analysis to determine titer for VLP 3 and VLP 1 variants.

	Densitometry Reaction 1 (μg/mL)	Densitometry Reaction 2 (μg/mL)	Dot Blot Reaction 1 (μg/mL)	Dot Blot Reaction 2 (μg/mL)
VLP 3	28.0 ± 0.9	30.9 ± 6.5	0.0 ^1^	24.5 ± 3.6
VLP 3 pJL1	39.4 ± 3.6	73.8 ± 8.3	52.1 ± 2.9	119.8 ± 2.4
VLP 3 pET28b no arg	35.8 ± 3.2	69.7 ± 23.1	0.0 ^1^	0.0 ^1^
VLP 3 pJL1 + linkers	95.1 ± 9.9	152.5 ± 4.6	124.8 ± 3.6	151.0 ± 8.5
VLP 1	14.6 ± 4.3	6.7 ± 1.0	17.6 ± 1.6	8.9 ± 1.5
VLP 1 pJL1	25.9 ± 4.3	19.7 ± 2.8	43.6 ± 2.7	34.3 ± 3.0
VLP 1 pET28b no arg	36.9 ± 2.1	27.4 ± 1.8	53.7 ± 2.0	97.3 ± 4.8
VLP 1 pJL1 + linkers	75.8 ± 4.7	51.0 ± 12.4	70.6 ± 1.6	76.8 ± 0.6

^1^ these samples were below the threshold of detection.

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
