# Peer review of "Escherichia coli-Based Cell-Free Protein Synthesis for Iterative Design of Tandem-Core Virus-Like Particles"

_vaccines, 2021, doi:10.3390/vaccines9030193_

Round 1

Reviewer 1 Report

General comment:

Thank you for the opportunity to review this manuscript. My recommendation is "Minor revision.". Colant et al developed an E. coli-based cell-free protein synthesis system to produce a variety of Virus-like particles.  Indeed, the authors reveal that this system was useful for the rapid synthesis of virus-like particle constructions. The manuscript has merits for readers; however, a bit of concern needs to be clarified before publication is addressed below.

Major comments:

  1. The introduction section is too long, therefore this section must be shortened.
  2. Figure 1D is hardly understood. Please consider replacing figure 1D with Figure A1.
  3. The results section is also too long, therefore this section must be shortened.
  4. The low molecular weight proteins that might be a result of partial transcription or translation are unavoidable or can be improved somehow? This must be discussed.

Author Response

We thank the reviewers for their thoughtful and detailed comments. We believe our responses have significantly improved the manuscript. The updates to the manuscript appear in red as tracked changes. These updates will be individually addressed in response to reviewer comments below.

1. The introduction section is too long, therefore this section must be shortened.

We have reduced the text in the introduction significantly. Please see the text that has been struck through on pages 1-5.

2. Figure 1D is hardly understood. Please consider replacing figure 1D with Figure A1.

We have removed Figure 1D from Figure 1 and created a new figure in the main text that was previosuly Figure A1 (now Figure 2).

3. The results section is also too long, therefore this section must be shortened.

We have reduced the text in the results section as well. Please see the text that has been struck through on pages 8-12. Some sentences have also been shortened or reworded rather than completely omitted. 

4. The low molecular weight proteins that might be a result of partial transcription or translation are unavoidable or can be improved somehow? This must be discussed.

We have included an explanation on page 20. Total removal of the lower molecular weight product related impurities seems very unlikely. But we can improve the ratio of full length product to impurities by adjusting the design of the vaccine. Here we suggest that removing the arginine rich region and adjusting the glycine-glycine-serine linkers may improve that ratio. However, those adjustments also impact particle formation. 

Reviewer 2 Report

In this manuscript, an E. coli-based cell-free protein synthesis (CFPS) platform was used to produce universal influenza vaccine candidates, by use of HBV core antigen (HBcAg) virus-like particles (VLPs) to display different antigens on its surface. The study sounds very interesting.

Here some concerns about this study:

  1. While the insolubility of the vaccine candidates VLP1 and VLP3 impact the inducing of immune response after vaccination?
  2. Following the above question, has any experiment been performed to evaluate the vaccine candidates capability to induce humoral immune response?

One minor concern:

  1. The VLP1 has been described being partial soluble if synthesized at lower temperature, perhaps a non-denaturing gel assessment would be helpful to further characterize the structure of it.

Author Response

We thank the reviewers for their thoughtful and detailed comments. We believe our responses have significantly improved the manuscript. The updates to the manuscript appear in red as tracked changes. These updates will be individually addressed in response to reviewer comments below.

1. While the insolubility of the vaccine candidates VLP1 and VLP3 impact the inducing of immune response after vaccination?

Yes. The vaccine product must be soluble before it can be injected. There are a few options for resolving the solubility of a vaccine product: reengineering the construct / changing cell free synthesis conditions or disassembling and reassembling in downstream processing. We have addressed this concern on page 17.

2. Following the above question, has any experiment been performed to evaluate the vaccine candidates capability to induce humoral immune response?

These vaccines have been shown to induce an immune response in mice (Ramirez et al. 2018) but have not yet been tested in humans. This has been noted on page 3. 

3. The VLP1 has been described being partial soluble if synthesized at lower temperature, perhaps a non-denaturing gel assessment would be helpful to further characterize the structure of it.

We agree that a non-denaturing gel would be beneficial for determining or comparing the native molecular weights. We have attempted to use denaturing gels in the past to examine these VLPs. Unfortunately, even after the ammonium sulfate precipitation, the sample still contains a large amount of host cell proteins (and very little VLP). We were unable to identify a singular band on the gel that would correlate to the VLP.